# Comparative In Vitro and In Vivo Evaluation of Anti-CCR8 Full-Sized IgG and Its Fab Fragments in Murine Colorectal Cancer Models

**DOI:** 10.3390/molecules30224445

**Published:** 2025-11-18

**Authors:** Tongshuo Hu, Rubin Jiao, Kevin J. H. Allen, Connor Frank, Mackenzie E. Malo, Ekaterina Dadachova

**Affiliations:** College of Pharmacy and Nutrition, University of Saskatchewan, Saskatoon, SK S7N 5E5, Canada; tbn310@mail.usask.ca (T.H.); ruj501@mail.usask.ca (R.J.); kja782@usask.ca (K.J.H.A.); csf876@mail.usask.ca (C.F.); mem510@mail.usask.ca (M.E.M.)

**Keywords:** CCR8+ regulatory T cells, MC38 colorectal cancer model, CT26 colorectal cancer model, whole anti-CCR8 IgG, anti-CCR8 Fab fragments, 111Indium, microSPECT/CT, pharmacokinetics

## Abstract

CCR8 chemokine receptor is a selective marker of tumor-infiltrating regulatory T cells (ti-Tregs) which interfere with the efficacy of checkpoint inhibitor immunotherapy (ICI) in many types of cancer. Eliminating CCR8+ ti-Tregs dramatically improves the results of subsequent ICI. We have recently reported using 225Actinium-labeled anti-CCR8 IgG for killing CCR8+ ti-Tregs in murine colorectal tumors which synergized with subsequent anti-CTLA4 ICI. Here, we aimed to compare the in vivo behavior of anti-CCR8 full-sized IgG and its Fab fragments to select the best antibody format for the pre-clinical development of this combination modality. Anti-CCR8 Fab fragments were generated by papain digest of the whole IgG. The whole IgG and Fab were conjugated to bifunctional chelating agent DOTA and labeled with 111Indium (^111^In). MC8 and CT6 murine colorectal tumor-bearing C57Bl6 and Balb/c mice, respectively, were administered ^111^In-DOTA-IgG or ^111^In-DOTA-Fab and imaged with microSPECT/CT at 2–72 h post-injection. A biodistribution was performed after the last imaging time point. Both ^111^In-DOTA-IgG and ^111^In-DOTA-Fab demonstrated high tumor uptake in both MC38 and CT26 tumors, with ^111^In-DOTA-IgG uptake being significantly higher from the 24 h time point and onwards. ^111^In-DOTA-Fab also exhibited pronounced kidney uptake which persisted even at 72 h. The kidney clearance and retention of ^111^In-DOTA-Fab might represent a problem during therapy employing 225Actimium or other long-lived therapeutic radionuclides by potentially causing a dose-limiting kidney toxicity. This imaging/biodistribution evaluation not only determined that full-size anti-CCR8 IgG is the optimal antibody format for pre-clinical development but also informed on the timing of immunotherapy administration in future radioimmunotherapy and immunotherapy combination studies.

## 1. Introduction

Regulatory T cells (Tregs) are an immunosuppressive subset of T lymphocytes which make an important contribution to immune homeostasis and self-tolerance [1]. Tumor-infiltrating Tregs (ti-Tregs), which are part of tumor microenvironment (TME), have been reported to contribute to the suppression of anti-tumor immune responses in the TME in several cancers [2]. Importantly, Tregs can decrease the efficacy of immunotherapy which creates the need for novel and precise strategies to target ti-Tregs [3]. Recently, the transcriptome of ti-Tregs in several cancers including colorectal cancer has been reported, which demonstrated the consistent upregulation of the CCR8 gene [4]. The CC chemokine receptor CCR8 is a seven transmembrane G-protein coupled receptor (GPCR) with a high affinity for human/mouse CCL1, mouse CCL8 (mCCL8), and human CCL18 (hCCL18), the latter being a functional analog of mCCL8 [5]. Thus, CCR8 is a biomarker of ti-Tregs which could make it possible to selectively target this subset of cells for depletion and to avoid killing the rest of Tregs. It has been demonstrated that while CCR8 blockade alone without simultaneous ti-Treg depletion was not sufficient to show antitumor effects, depletion of ti-Tregs by natural killer cells reduced the tumor growth and displayed synergy with anti-PD-1 therapy, producing tumor remission and generating immunological memory [6]. However, cellular therapies of cancer so far have been very logistically complex and expensive, putting them out of reach for many patients.

We have recently demonstrated that an anti-CCR8 antibody armed with alpha particles emitter 225Actinium (^225^Ac) was able to efficiently kill CCR8+ ti-Tregs in CT26 and MC38 murine models of colorectal cancer [7]. Moreover, when CT26 and MC38 tumor-bearing mice were treated with ^225^Ac-labeled anti-CCR8 antibody followed by anti-CTLA4 immune-checkpoint-inhibiting immunotherapy, the synergistic slowing down of tumor progression and increase in animal survival were observed [7]. As alternative antibodies formats such as Fab fragments could have potential advantages for delivering radionuclides to the tumors due to their easier tumor penetration and fast clearance from the circulation, in this study we generated Fab fragments of the same anti-CCR8 antibody used in [7] and evaluated their pharmacokinetics side by side with the whole anti-CCR8 IgG in CT26 and MC38 murine colorectal cancer models.

## 2. Results

### 2.1. Structural Analyses of Anti-CCR8 Whole IgG and Its Fab Fragments

Fab fragments were generated by papain digest of the whole anti-CCR8 IgG. Figure 1A shows the high-performance liquid chromatography (HPLC) profile of Fab on the size exclusion column with the peak eluting at 7.5 min. Figure 1B displays the SDS-PAGE under reducing conditions of whole IgG (lane 2) and Fab fragments (lane 3). The major band for the whole IgG is at 50 kDa and for Fab, at 25 kDa, which confirms their molecular identity.

### 2.2. Conjugation of Whole Anti-CCR8 IgG and Fab Fragments to DOTA and Their Immunoreactivity

To enable radiolabeling, the antibody and Fab fragments were conjugated to the bifunctional chelator DOTA (S-2-(4-Isothiocyanatobenzyl)-1,4,7,10-tetraazacyclododecane tetraacetic acid using 20:1 initial molar excess of DOTA over antibodies. The resulting chelator-to-antibody ratio (CAR) was determined by matrix-assisted laser desorption ionization mass spectrometry (MALDI MS) to be 3.5 for IgG and 2.9 for Fab fragments.

### 2.3. Immunoreactivity of Whole Anti-CCR8 IgG and Fab Fragments Towards CCR8 Before and After Conjugation to DOTA

Flow cytometry experiments using CCR8+ JC65 cell line and CCR8- Jurkat-NFAT-Luc-FcyRIII cell line revealed CCR8-specific binding for both whole IgG and its Fab fragments (Figure 2). Importantly, no loss of immunoreactivity for both whole anti-CCR8 IgG and its Fab fragments post-conjugation to DOTA was observed (Figure 2).

### 2.4. Both DOTA-Conjugated Anti-CCR8 Whole IgG and Its Fab Fragment Radiolabeled Quantitively with ^111^In

We then proceeded to radiolabeling the Fab fragments and whole anti-CCR8 IgG with an imaging radionuclide 111Indium (^111^In) for the subsequent in vivo imaging in colorectal cancer murine models. The radiolabeling yields were >95% as determined by instant thin layer chromatography (iTLC) and HPLC (Figure 3) and no purification was required before in vivo experiments.

### 2.5. Microspect/CT Imaging and Biodistribution of ^111^In-Labeled Anti-CCR8 IgG and Fab Fragments in Murine Colorectal Tumor Models CT26 and MC38

CT26 and MC38 murine colorectal tumors were initiated in female Balb/c and C57Bl6 mice, respectively, and when the tumors reached a volume of ~200 mm^3^, the mice were randomized into the groups of four animals, and injected into the tail vein with either 200 µCi anti-CCR8 ^111^In-IgG or ^111^In-Fab. The mice were imaged on MILabs SPECT/PET/CT camera at 2, 4, 24, 48, and 72 h, then humanely sacrificed at 72 h after the imaging and the biodistribution were performed. Figure 4A shows the microSPECT/CT images of ^111^In-IgG and ^111^In-Fab in MC38 tumor-bearing C57Bl6 female mice. The images as well as quantification of tumor uptake as standardized uptake values (SUVs) (Figure 4B) revealed the following: (1) the tumor uptake of ^111^In-IgG was slightly lower than that of ^111^In-Fab at 2 and 4 h time points and then approximately 1.4 times higher than that of ^111^In-Fab for 24, 48, and 72 h period; (2) while ^111^In-IgG was primarily localizing in the tumor and in the liver following the hepatobiliary excretion route—^111^In-Fab fragments were demonstrating high uptake in the kidneys; (3) for both ^111^In-IgG and ^111^In-Fab fragments, the highest uptake in the tumor was observed at 24 h post-administration. Figure 5 displays the tumor-to-organ ratios for ^111^In-IgG (Figure 5A) and ^111^In-Fab (Figure 5B). The tumor-to-organ ratios which exceeded the value of 1 were observed for the kidneys and heart for ^111^In-IgG and only for the heart for ^111^In-Fab. ^111^In-IgG time–activity curves revealed steady decrease in activity in major organs starting from the 2 h time point and accumulation of activity in the tumor up to 24 h (Figure 5A) while ^111^In-Fab activity peaked in the kidneys at 4 h, steadily decreased in the liver and heart, and stayed almost constant in the tumor with slight maximum at 24 h, after which it started to decrease (Figure 5B).

Similar trends were observed when CT26 tumor-bearing female Balb/c mice were administered ^111^In-IgG and ^111^In-Fab and imaged with microSPECT/CT (Figure 6A). As CT26 tumors are less aggressive and are more uniform in size, the difference in tumor SUVs between ^111^In-IgG and ^111^In-Fab reached statistical significance at 24, 48, and 72 h post-administration (Figure 6B). Figure 7 displays the tumor-to-organ ratios for ^111^In-IgG (Figure 7A) and ^111^In-Fab (Figure 7B). The tumor-to-organ ratios which exceeded the value of 1 were observed for the kidneys and heart for ^111^In-IgG and for the heart and liver for ^111^In-Fab, while the time–activity curves for ^111^In-IgG and ^111^In-Fab were quite similar to those in the MC38 model.

After the final imaging time point at 72 h, the mice were humanely sacrificed and the biodistribution was performed (Figure 8). The biodistribution results confirmed high kidney uptake of ^111^In-Fab in both models with uptake in C57Bl6 mice (Figure 8A) being higher than in Balb/c mice (Figure 8B). Tumor uptake of anti-CCR8 ^111^In—IgG in both MC38 and CT26 tumors was ≥15% ID/g, which should translate into significant tumoricidal dose during radioimmunotherapy.

## 3. Discussion

CCR8 has emerged as a biomarker for tumor-infiltrating Tregs [5,6] which interfere with the action immune checkpoint inhibitors. We have recently demonstrated for the first time that an anti-CCR8 antibody armed with alpha particles emitter ^225^Ac was able to efficiently kill CCR8+ ti-Tregs in CT26 and MC38 murine models of colorectal cancer [7]. In this current study, we generated Fab fragments of the same anti-CCR8 IgG used in [7] and compared their pharmacokinetics with the whole IgG in CT26 and MC38 murine colorectal cancer models.

The motivation for this study was provided by the fact that hematologic toxicity is a major limitation in radioimmunotherapy with intact IgG antibodies (molecular weight ~150 kDa) and constitutes the primary dose-limiting factor. This toxicity arises from full antibodies prolonged circulation, leading to extended systemic exposure and irradiation of the bone marrow [8]. Dose fractionation and/or the use of smaller antibody derivatives, such as F(ab′)_2_ fragments (~100 kDa) or Fab fragments (~50 kDa), can mitigate bone marrow toxicity due to their rapid clearance via renal excretion, although this strategy may concomitantly elevate the risk of nephrotoxicity [9]. In parallel, there is increasing emphasis on theranostic approaches, wherein gamma-emitting analogs of therapeutic radiopharmaceuticals are employed for tumor visualization, assessment of target expression, and estimation of absorbed radiation doses to both tumors and normal tissues prior to alpha- or beta-particle therapy. Antibody fragments, by virtue of yielding higher tumor-to-blood and tumor-to-normal tissue ratios compared to intact IgG, enhance imaging quality and may therefore provide distinct advantages for theranostic applications [10]. In this regard, Kondo et al. have recently reported generation and evaluation of the ^225^Ac-labeled trastuzumab IgG, F(ab’)_2_ and Fab for theranostic SPECT/CT imaging and alpha-particle radioimmunotherapy of HER2-positive human breast cancer [11].

The challenge in making the Fab fragments of the anti-CCR8 antibody used in our study is its IgG2b isotype. It has been shown that among the common IgG isotypes, such as IgG1, IgG2a, and IgG2b, the latter is the most sensitive to the pH of the buffer, time and temperature of incubation, and to the amount of enzymes added during the generation of F(ab)_2_ and Fab fragments [12,13,14], which might result in relatively low yields of the fragments. In spite of this challenge, we were able to generate enough Fab fragments to perform their side-by-side comparison with anti-CCR8 whole IgG in vitro and in vivo.

The in vitro behavior of both whole IgG and Fab fragments revealed their specific binding to CCR8 receptor on CCR8+ JC65 cells and preservation of this immunoreactivity towards CCR8 post-conjugation to the bifunctional chelating agent DOTA. CARs of 2.9 and 3.5 for Fab fragments and whole IgG, respectively, resulted in quantitative radiolabeling with ^111^In. In our experience, CARs between 3 and 4 are sufficient for quantitative radiolabeling of antibodies with trivalent radiometals while not perturbing the antibodies binding to their respective antigens [15,16].

The in vivo behavior of whole anti-CCR8 IgG and its fragments was compared in two diverse models of murine colorectal cancer—MC38 tumors in C57Bl6 mice and CT26 tumors in Balb/c mice. In both models, the tumor uptake of ^111^In-labeled IgG and Fab fragments was slightly higher for Fab or equal to that of IgG for the early time points of 2 and 4 h post-administration, and became 1.4–1.5 higher for whole IgG from the 24 h time point and onwards. The highest tumor uptake for whole IgG and Fab fragments in both models was achieved at 24 h post-administration which may have important implications for the subsequent timing of administering immune checkpoint inhibitors. Also, the clearance patterns of the whole IgG and Fab fragments were similar between two models—hepatobiliary route for IgG and kidney excretion for Fab fragments. Some differences in the absolute values of kidney and other normal organs’ uptake between the two models were obviously related to mouse strain. The kidney uptake of anti-CCR8 Fab fragments was very high throughout the 72 h terminal time point, which indicates that if the fragments are labeled with a long-lived therapeutic radionuclide such as 225Actinium or 177Lutetium, kidneys might become a dose-limiting organ.

This high kidney uptake of anti-CCR8 Fab fragments is due to their molecular weight of approximately 50 kDa. Antibody fragments with a molecular weight below the ~60 kDa glomerular filtration threshold are cleared faster than whole IgGs, especially since they lack an Fc region which would allow recycling via the FcRn pathway [17]. Because of their smaller size, Fab fragments undergo glomerular filtration and are subsequently reabsorbed into the renal cells. Afterwards, the lysosomal degradation occurs to those low-molecular-weight antibody fragments, releasing associated metabolites (such as amino acids and their radiolabeled derivatives); those major metabolites are quickly excreted, while DOTA-radiometal amino acids (such as ^111^In-, ^88^/^90^Y- or ^161^Tb-DTPA-lysine) remain trapped, leading to persistent kidney retention and irradiation of kidney cells with beta or alpha particles, resulting in toxicity [18]. A potential solution to this problem of high kidney uptake could be the implementation of linkers cleavable by the renal brush border membrane enzymes, which could decrease the radioactive doses to the kidney without impairing the tumor uptake [19]. In addition, positively charged amino acids such as arginine or lysine were utilized in the past [18], while more recent approaches include co-administration of sodium paraaminohippurate with various radiopharmaceuticals to reduce their kidney uptake [20].

Overall, two important conclusions influencing the future combination of radioimmunotherapy targeting CCR8+ ti-Tregs and immune checkpoint inhibitors can be derived from these experiments: (1) whole anti-CCR8 IgG shows higher uptake in both CT26 and MC38 tumors and no uptake in the kidneys and should be used in combination with anti-CTLA4 and anti-PD1 immunotherapy; (2) the highest uptake of the antibody in the tumor is reached at 24 h post-injection. Thus, the administration of immunotherapy should happen after RIT at 24–48 h post RIT administration and not simultaneously as performed in our previous study [7]. Allowing the radiolabeled antibody to reach its highest uptake in the tumor before giving immunotherapy will allow for more profound killing of CCR8+ Tregs in the tumor before CD8+ T cells and other immune cells will start coming into the tumor microenvironment [21].

## 4. Materials and Methods

### 4.1. Generation of Fab Fragments

To generate Fab fragments from anti-CCR8 antibody, the purified anti-mouse CD198 (CCR8) Antibody (BioLegend, San Diego, CA, USA, Cat#96198) was exchanged using a Amicon^®^ Ultra Centrifugal Filter, 30 kDa MWCO (Sigma-Aldrich, St Louis, MO, USA, Ref#503096) into a combination buffer (20 mM NaH_2_PO_4_, Fisher Chemical, Pittsburgh, PA, USA, Cat#S369-1), 10 mM EDTA (UltraPure™ 0.5M EDTA, pH 8.0, Invitrogen, Eugene, OR, USA, Ref#15575-038) with pH = 7.0. Three hundred µL of the combination buffer was added to the antibody and centrifuged at 14,000× *g* for 3 min. This step was repeated ten times, follow by inversion of the ultrafilter and centrifugation at 500× *g* for 3 min to recover the buffer-exchanged IgG. The concentration of the IgG was measured using NanoDrop One/Nanodrop One C. Meanwhile, 2.5% immobilized Papain (Agarose Resin, Thermo Scientific, Waltham, MA, USA) was freshly prepared by equilibrating it in digestion buffer consisting of the combination buffer and 80 mM L-cysteine (L-cystein, Sigma-Aldrich, Cat#168149-25G) at a 1:4 *v*/*v* ratio. Afterward, the equilibrated papain was added to the buffer-exchanged antibody, purged with N_2_, and incubated at 37 °C for 20 h with rotation in HB-1000 Hybridizer, thus creating the Fab fragments crude digest. The reaction was stopped by adding 10 mM Tris-HCl (Tris(hydroxymethyl)aminomethane, Acros Organics, Waltham, MA, USA, Cat# 424570025), pH = 7.5, and Fab fragments were clarified by using Protein G MagBeads (GenScript, Piscataway, NJ, USA, Cat#L00274). For this 10 µL of the protein G per 25 µg of antibody were utilized. The beads were washed with washing buffer (20 mM Na_2_HPO_4_, 0.15 M NaCl) with pH = 7.02. EasySep™ Magnet (Stemcell Technologies, Cambridge, MA, USA, Cat#18000) was used to separate the beads and supernatant; the supernatant was removed and washing was repeated 3 times. The crude digest was added to the washed protein G and shaken at room temperature for 60 min in an orbital mixing chilling/heating plate (Torrey Pines Scientific, San Marcos, CA, USA). The supernatant was collected and buffer-exchanged into the conjugation buffer for conjugation of Fab to the chelating agent DOTA.

### 4.2. Conjugation of Whole IgG and Fab Fragments to DOTA, CAR Determination by MALDI MS and Radiolabeling of DOTA-Conjugated Whole IgG and Fab Fragments with ^111^In

The whole anti-CCR8 IgG and its Fab fragments were conjugated to bifunctional chelating agent DOTA (Macrocyclics, Plano, TX, USA) using 20 molar initial excess of DOTA over the antibodies as described in [7]. The final CARs were determined at the University of Alberta, Canada, MS Facility. Radiolabeling of the DOTA-conjugated antibodies with ^111^In was performed as in [7]. The radiolabeling yields for both whole IgG and Fab fragments were >95% and they were used for in vivo experiments without purification.

### 4.3. Flow Cytometry of Whole IgG and Fab Fragments Binding to CCR8+ T Cells

Flow cytometry was performed to verify the binding of anti-CCR8 fragments to CCR8. JC65 (CCR8+ cell line) and Jurkat-NFAT-Luc-FcyRIII (CCR8− cell line) cells were grown in RPMI 1640 + 10% FBS at 5% CO_2_. Cells were centrifuged at 300× *g* with 9 acceleration and 9 deacceleration for 5 min at 4 °C; the supernatant was removed and the cell concentration was adjusted to 2 × 10^6^ cells/mL by resuspending them in the FACS buffer (PBS + 2% FBS + 0.02% Sodium Azide). In the flow panel, 100 µL of the resuspended cells was added to each well. For each cell line, there were four different groups: (1) naïve anti-CCR8 IgG, (2) DOTA-conjugated anti-CCR8 IgG, (3) DOTA-conjugated anti-CCR8 Fab fragments, and (4) no primary antibodies. The concentration of the primary antibodies was 5 µg/mL; they were incubated for 60 min at 4 °C without light. After the incubation of primary antibodies, the flow panel was centrifuged at 500× *g* with 9 accelerations and 9 deaccelerations for 3 min at 4 °C, supernatant was removed, and the cells were resuspended in FACS buffer; this washing step was repeated 3 times. Two different secondary antibodies were used for each of the primary antibody groups: (1) PE anti-rat Ig light chain κ Antibody (BioLegend, cat#407806); the staining with this secondary antibody was performed at 1:6700 dilution in FACS buffer. (2) Mouse anti-Rat IgG2b Secondary Antibody, PE (Invitrogen, Cat#12-4815-82); the staining with this secondary antibody was performed at a 1:1000 dilution in FACS buffer. The cells were incubated for 30 min on ice without light. Afterward, cells were washed three times and resuspended in FACS buffer. Beckman Coulter CytoFlex flow cytometer (Indianapolis, IN, USA) was used for analyses. Data was processed and analyzed using FlowJo software (Version 10.4.).

### 4.4. SPECT/CT Imaging of CT26 and MC38 Tumor-Bearing Mice with ^111^In-Labeled Anti-CCR8 IgG and Its Fab Fragments

CT26 and MC38 murine colorectal cancer cells (ATCC, Mannassas, VA, USA) were grown as in [7]. CT26 and MC38 tumors were initiated in female Balb/c and C57Bl6 mice, respectively, as in [7] and when the tumors reached ~200 mm^3^ volume, the mice were randomized into the groups of 4 animals, and equimolar amounts of either 200 µCi anti-CCR8 ^111^In-IgG or ^111^In-Fab were injected into the tail vein. In this regard, anti-CCR8 ^111^In-IgG was labeled at a 5:1 µCi:µg specific activity whereas the anti-CCR8 ^111^In-Fab was labeled at a 15:1 µCi:µg specific activity to ensure that an equimolar amount of antibody and Fab were injected (40 µg antibody and 13.3 µg Fab). The mice were imaged on MILabs SPECT/PET/CT camera at 2, 4, 24, 48, and 72 h. microSPECT/CT images are presented as MIP images and scaled between a low value (blue) of 0 kBq/cc and a high value (red) of 700 kBq/cc. Organ and tumor regions of interest (ROIs) were drawn manual in Slicer3D (Version 5.6.2) based on CT images and exported for analysis in pMOD (Version 3.910). The animals were humanely sacrificed at 72 h time point after the imaging and the biodistribution was performed. The image processing and SUV calculations were performed as in [22].

### 4.5. Statistical Analysis

Two-way ANOVA was used for both imaging and biodistribution data analyses, with the Sidak’s multiple comparison performed within the groups. *p* values < 0.05 were considered statistically significant.

## Figures and Tables

**Figure 1 molecules-30-04445-f001:**
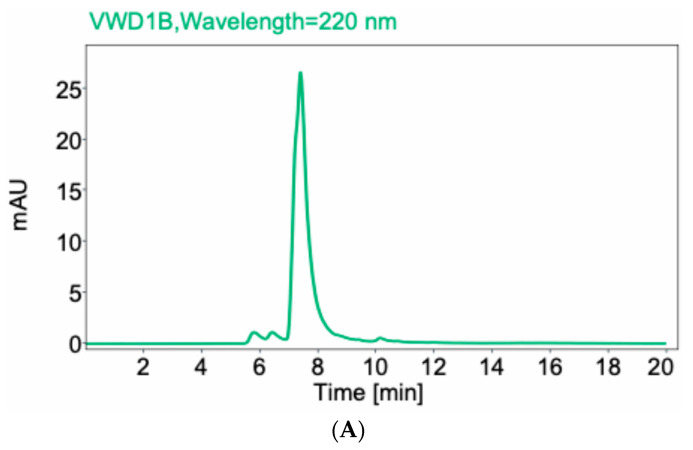
Size exclusion HPLC of anti-CCR8 Fab fragments (**A**) and SDS-PAGE under reducing conditions of whole anti-CCR8 IgG and its Fab fragments (**B**). 10% gel was used. Lane 1: protein ladder; Lane 2: whole anti-CCR8 antibody; Lane 3: anti-CCR8 Fab fragments; Lane 4: protein ladder; Lane 5: anti-CCR8 Fab conjugated to the bifunctional chelating agent DOTA.

**Figure 2 molecules-30-04445-f002:**
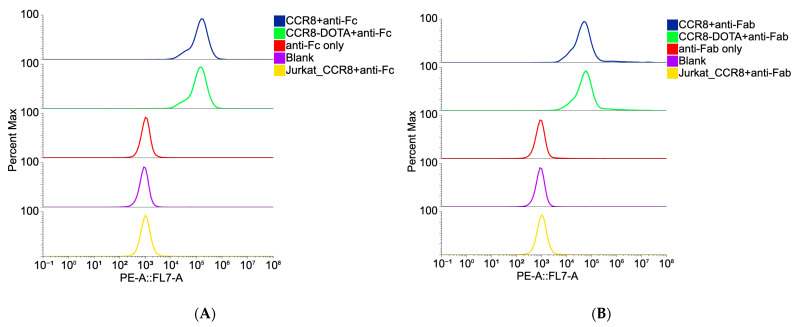
Binding of DOTA-conjugated and non-conjugated anti-CCR8 IgG (**A**) and its Fab fragments (**B**) to CCR8 + JC65 cells and CCR8-Jurkat-NFAT-Luc-FcyRIII cells by flow cytometry. Anti-Fc secondary antibody was used for whole IgG (**A**); anti-Fab secondary antibody was used for Fab fragments (**B**).

**Figure 3 molecules-30-04445-f003:**
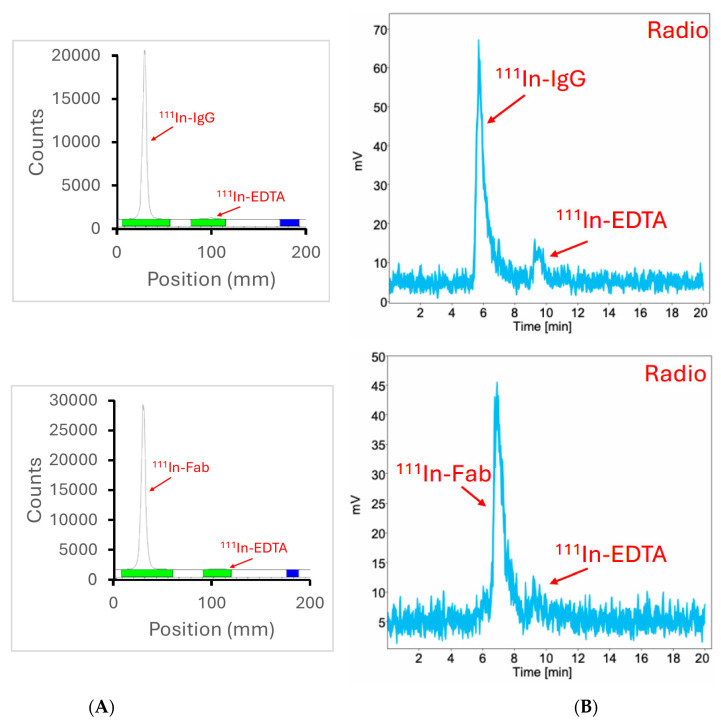
iTLC and radioHPLC of ^111^In-labeled anti-CCR8 IgG and Fab fragments: (**A**) iTLC of ^111^In- antibody is shown in the upper panel, of ^111^In-Fab—in the lower panel; (**B**) radioHPLC of ^111^In-antibody is shown in the upper panel, of ^111^In-Fab—in the lower panel.

**Figure 4 molecules-30-04445-f004:**
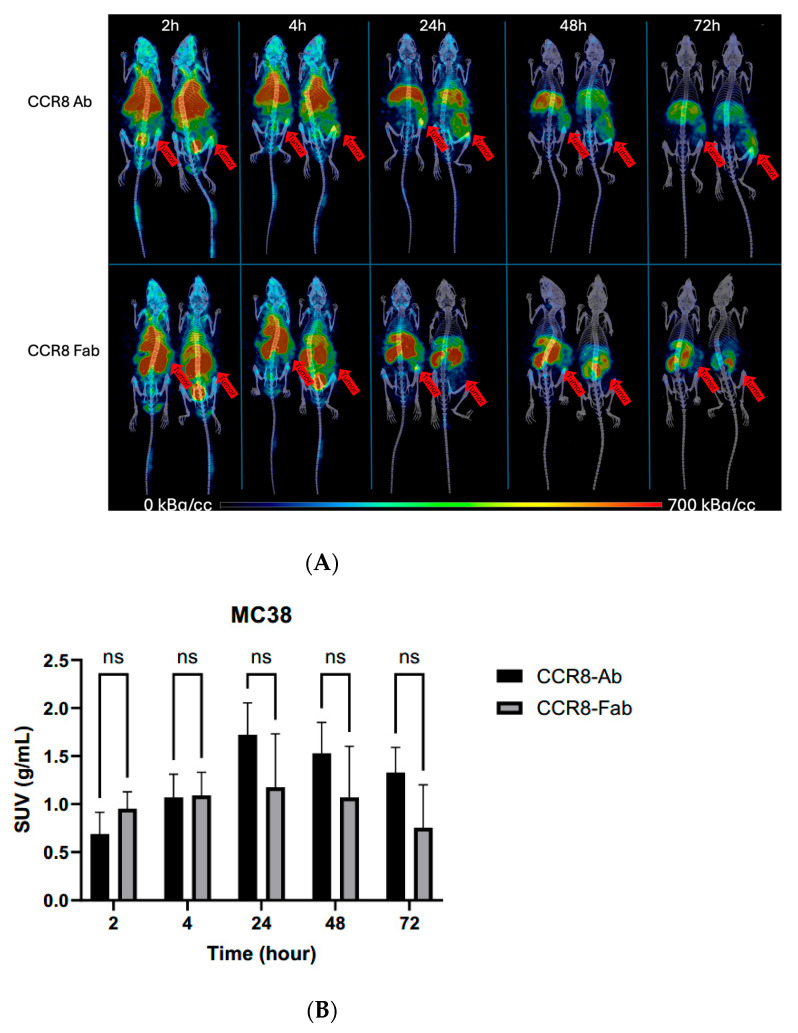
MicroSPECT/CT imaging of MC38 tumor-bearing C57Bl6 female mice injected with anti-CCR8 ^111^In-IgG and ^111^In-Fab fragments. The mice were injected IV and imaged at 2–72 h: (**A**) microSPECT/CT images of ^111^In-IgG (upper panel), and ^111^In-Fab fragments (lower panel). The plates show times from 2 to 72 h. Red arrows are pointing to the tumors. The red color on the images represent the highest activity, blue—the lowest; (**B**) tumor SUV for ^111^In-IgG and ^111^In-Fab fragments. ns—not significant statistical difference.

**Figure 5 molecules-30-04445-f005:**
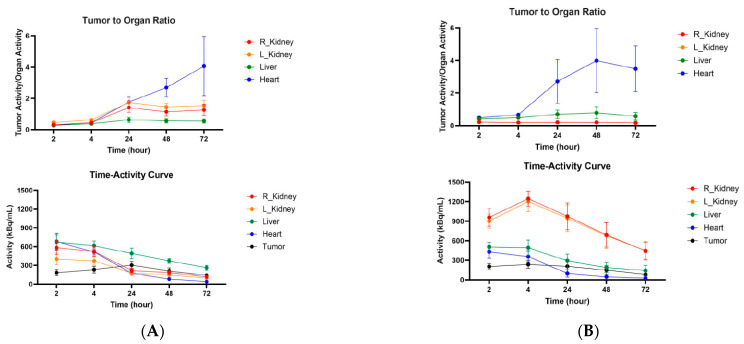
Tumor-to-organ ratios and time–activity curves for MC38 tumor-bearing C57Bl6 female mice injected with anti-CCR8 ^111^In-IgG and ^111^In-Fab fragments. The mice were injected IV and imaged at 2–72 h: (**A**) tumor-to-organ ratios (upper panel) and time–activity curves (lower panel) of ^111^In-IgG; (**B**) tumor-to-organ ratios (upper panel) and time–activity curves (lower panel) of ^111^In-Fab fragments.

**Figure 6 molecules-30-04445-f006:**
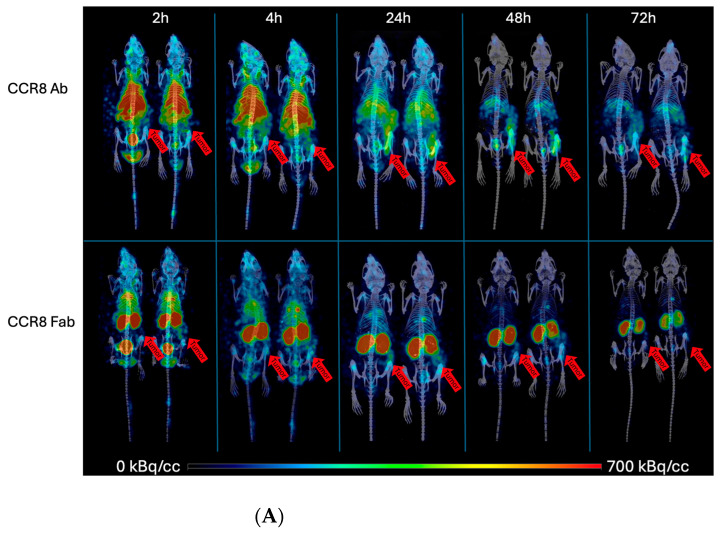
microSPECT/CT imaging of CT26 tumor-bearing Balb/c female mice with anti-CCR8 ^111^In-IgG and ^111^In-Fab fragments. The mice were injected IV and imaged at 2–72 h: (**A**) microSPECT/CT images of ^111^In-IgG (upper panel) and ^111^In-Fab fragments (lower panel). The plates show times from 2 to 72 h. Red arrows are pointing to the tumors. The red color on the images represent the highest activity, blue—the lowest; (**B**) tumor SUV for ^111^In-IgG and ^111^In-Fab fragments. *p* values < 0.001 are shown as three stars and *p* values < 0.0001 as four stars.

**Figure 7 molecules-30-04445-f007:**
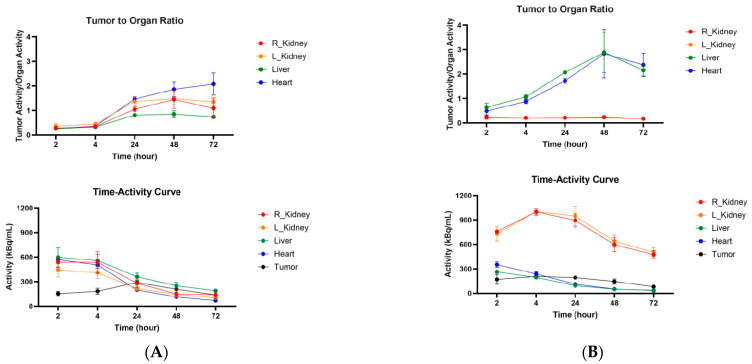
Tumor-to-organ ratios and time–activity curves for CT26 tumor-bearing Balb/c female mice injected with anti-CCR8 ^111^In-IgG and ^111^In-Fab fragments. The mice were injected IV and imaged at 2–72 h: (**A**) tumor-to-organ ratios (upper panel) and time–activity curves (lower panel) of ^111^In-IgG; (**B**) tumor-to-organ ratios (upper panel) and time–activity curves (lower panel) of ^111^In-Fab fragments.

**Figure 8 molecules-30-04445-f008:**
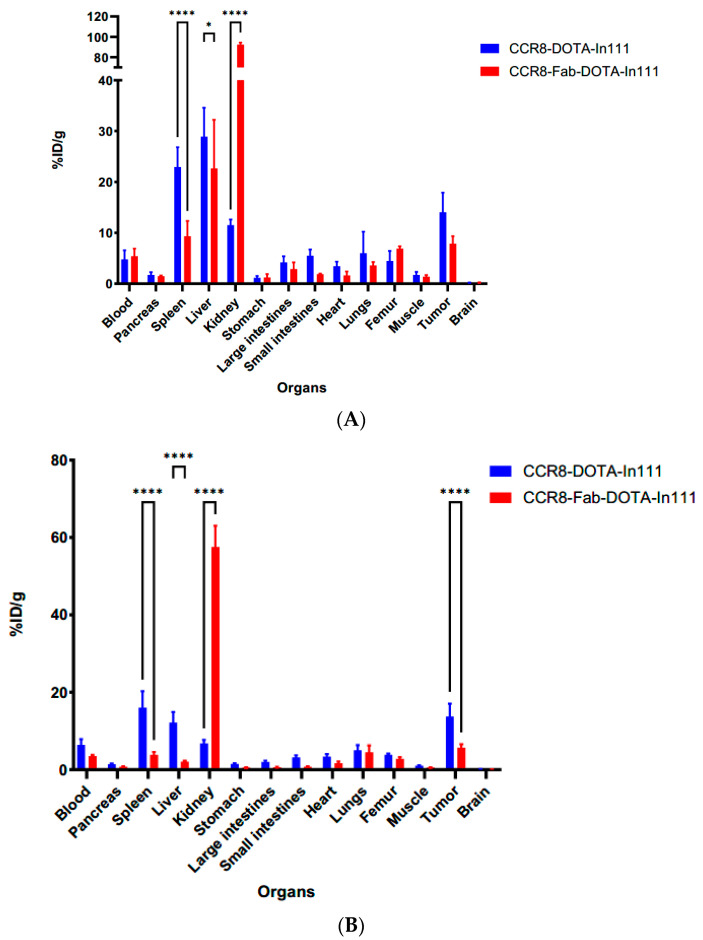
Biodistribution of anti-CCR8 ^111^In-IgG and ^111^In-Fab at 72 h post-IV-administration in MC38 tumor-bearing C57Bl6 mice (**A**) and CT26 tumor-bearing Balb/c mice (**B**). *p* values < 0.05 are shown as one star, and *p* values < 0.0001 as four stars.

## Data Availability

All data is included in the manuscript. The research protocols used in the study are available from the corresponding author upon request.

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
