# Peer review of "Comparative In Vitro and In Vivo Evaluation of Anti-CCR8 Full-Sized IgG and Its Fab Fragments in Murine Colorectal Cancer Models"

_molecules, 2025, doi:10.3390/molecules30224445_

Round 1

Reviewer 1 Report

Comments and Suggestions for Authors

This study compares the in vitro and in vivo effects of anti-CCR8 full-length IgG and its Fab' fragments in murine colorectal cancer models. Nevertheless, the manuscript presents several significant concerns:

(1) The study lacks in vivo efficacy evaluation, and the in vitro cytotoxicity assays are overly simplistic. Additional experiments are required to substantiate the findings.

(2) There is insufficient discussion regarding the underlying reasons for the differential toxicity profiles observed between the two formulations.

(3) The methodology used for determining in vivo biodistribution remains unclear and requires elaboration.

(4) The English expression throughout the manuscript needs substantial revision to meet standards of academic writing.

Author Response

Reviewer 1

This study compares the in vitro and in vivo effects of anti-CCR8 full-length IgG and its Fab fragments in murine colorectal cancer models. Nevertheless, the manuscript presents several significant concerns:

1)The study lacks in vivo efficacy evaluation, and the in vitro cytotoxicity assays are overly simplistic. Additional experiments are required to substantiate the findings. - Response: The goal of this study was to perform the selection of the best anti-CCR8 antibody format for future radioimmunotherapy + ICI combination therapy, this is why we did not perform in vitro or in vivo toxicity evaluation in this study. These will be done in the near future.  To clarify this, we have now added a following sentence to the abstract: “This imaging/biodistribution evaluation not only determined that full size anti-CCR8 IgG is the optimal antibody format for pre-clinical development but also informed on the timing of immunotherapy administration in the future radioimmunotherapy and immunotherapy combination studies.”

2) There is insufficient discussion regarding the underlying reasons for the differential toxicity profiles observed between the two formulations.  – Response: We have now expanded the section in the Discussion on the potential kidney toxicity of the fragments and added new ref. 20. It now reads: “This high kidney uptake of anti-CCR8 Fab fragments is due to their molecular weight of  approximately 50 kDa. Antibody fragments with a molecular weight below the ~60-kDa glomerular filtration threshold are cleared faster than whole IgGs, especially since they lack an Fc region which would allow recycling via the FcRn pathway [17].  Because of their smaller size, Fab fragments undergo glomerular filtration and are subsequently reabsorbed into the renal cells. Afterwards, the lysosomal degradation occurs to those low molecular weight antibody fragments, releasing associated metabolites (such as amino acids and their radiolabeled derivatives; those major metabolites are quickly excreted, while DOTA-radiometal amino acids (such as 111In-, 88/90Y- or 161Tb-DTPA-lysine) remain trapped, leading to persistent kidney retention and irradiation of kidney cells with beta- or alpha-particles resulting in toxicity [18]. Potential solution to this problem of high kidney uptake could be implementation of linkers cleavable by the renal brush border membrane enzymes which could decrease the radioactive doses to the kidney without impairing the tumor uptake [19]. In addition, positively charged aminoacids such as arginine or lysine were utilized in the past [18] while more recent approaches include co-administration of sodium paraaminohippurate with various radiopharmaceuticals to reduce their kidney uptake [20].

(3) The methodology used for determining in vivo biodistribution remains unclear and requires elaboration. – Response: We have now expanded Section 4.4 in the Materials and Methods. The following information has been added: “microSPECT/CT images are presented as MIP images and scaled between a low value (blue) of 0 kBq/cc and a high value (red) of 700 kBq/cc. Organ and tumor regions of interest (ROIs) were drawn manual in Slicer3D (v .5.6.2) based on CT images and exported for analysis in pMOD (v. 3.910).”

(4) The English expression throughout the manuscript needs substantial revision to meet standards of academic writing. – Response: We have now performed an extensive revision of the English language throughout the manuscript.

Reviewer 2 Report

Comments and Suggestions for Authors

The manuscript presents a comparative evaluation of full-sized anti-CCR8 IgG and its Fab’ fragments, focusing on their biodistribution and pharmacokinetics in two murine colorectal cancer models (MC38 and CT26). The study extends prior work by the same group that explored CCR8-targeting radioimmunotherapy, aiming to identify the optimal antibody format (IgG vs. Fab’) for preclinical development.

The study is well performed and convincing, demonstrating higher and more sustained tumor retention of the full IgG compared to Fab’, as well as problematic renal accumulation of Fab’. The authors’ conclusion that the IgG format should be prioritized for future CCR8-targeting radioimmunotherapy is justified.

However, the main issue with this study is that it investigated whether Fab’ fragments would improve the anti-CCR8 IgG protocol based on the assumption that smaller molecules would be at least equally efficient and less toxic than the whole IgG antibody. This was not demonstrated; therefore, the paper essentially presents negative results and does not advance the previous work (see ref. 7).

Author Response

Reviewer 2

The manuscript presents a comparative evaluation of full-sized anti-CCR8 IgG and its Fab fragments, focusing on their biodistribution and pharmacokinetics in two murine colorectal cancer models (MC38 and CT26). The study extends prior work by the same group that explored CCR8-targeting radioimmunotherapy, aiming to identify the optimal antibody format (IgG vs. Fab) for preclinical development. The study is well performed and convincing, demonstrating higher and more sustained tumor retention of the full IgG compared to Fab, as well as problematic renal accumulation of Fab. The authors’ conclusion that the IgG format should be prioritized for future CCR8-targeting radioimmunotherapy is justified.

However, the main issue with this study is that it investigated whether Fab fragments would improve the anti-CCR8 IgG protocol based on the assumption that smaller molecules would be at least equally efficient and less toxic than the whole IgG antibody. This was not demonstrated; therefore, the paper essentially presents negative results and does not advance the previous work (see ref. 7). – Response: We would like to thank the Reviewer for their high opinion of our work, however, we respectfully disagree that this study has not advanced the previous work.  First of all, there is no published information on targeting ti-Tregs with any form of a radiolabeled antibody, and thus comparing a full antibody to its Fab fragments was performed for the first time. And our imaging study also informed us on an optimal time of administration of immunotherapy in combination studies. In this regard, our original manuscript states: “Overall, two important conclusions influencing the future combination of radioimmunotherapy targeting CCR8+ ti-Tregs and immune checkpoint inhibitors can be derived from these experiments: 1) whole anti-CCR8 IgG shows higher uptake in both CT26 and MC38 tumors and no uptake in the kidneys and should be used in combination with anti-CTLA4 and anti-PD1 immunotherapy; 2) the highest uptake of the antibody in the tumor is reached at 24 hrs post injection. Thus, the administration of immunotherapy should happen after RIT at 24-48 hrs post RIT administration and not simultaneously as it was done in our previous study [7]. Allowing the radiolabeled antibody to reach its highest uptake in the tumor before giving immunotherapy will allow for more profound killing of CCR8+ Tregs in the tumor before CD8+ T cells and other immune cells will start coming into the tumor microenvironment [21].”  To emphasize the importance of the 2nd point, we have now added a sentence to the Abstract which reads: “This imaging/biodistribution evaluation not only determined that full size anti-CCR8 IgG is the optimal antibody format for pre-clinical development but also informed on the timing of immunotherapy administration in the future radioimmunotherapy and immunotherapy combination studies.”

Reviewer 3 Report

Comments and Suggestions for Authors

The authors compare full-length anti-CCR8 IgG and its Fab’ fragment, each DOTA-conjugated and ¹¹¹In-radiolabeled, in two syngeneic colorectal-cancer mouse models. The study addresses an important translational step - selecting an optimal antibody format for CCR8-directed theranostics.  The experiments are clear and technically sound, yet several aspects require clarification or additional analysis before publication.
1. The claim that whole IgG is “optimal” would be stronger with absorbed-dose or time-integrated-activity estimates for tumor, marrow, and kidney.  Using the existing SUV/biodistribution data, please provide basic dosimetric projections (MBq·h/g or Gy/MBq) or at least time–activity curves and tumor-to-organ ratios over time.
2. It'd be better to report exact injected protein mass (µg or pmol) for both formats.  Because Fab’ and IgG differ markedly in molecular weight and Fc-mediated recycling, matching by radioactivity alone can misrepresent molar equivalence.
3. 
Given the persistent Fab’ renal uptake, it would be valuable to test or cite data for coinfusion with lysine/gelofusine or a cleavable linker.  Alternatively, re-frame the conclusion as “current Fab’ format limited by kidney retention.”
4. Add scale bars and color-scale legends to SPECT/CT images; ensure consistent tumor ROI definition. Plot %ID/g and tumor-to-organ ratios alongside SUVs for clarity.
5. Please compare these findings with other format-comparison studies (e.g., HER2-targeted IgG vs Fab’ RIT) to place the work in a broader theranostic context in discussion.
6. Correct
typographical errors (“highes activity” → highest activity; “immune chedck point” → checkpoint; “radiolabelieng” → radiolabeling).
7. I
mprove figure font size and label legibility, like Figure 1A, Figure 3.

Author Response

Reviewer 3

The authors compare full-length anti-CCR8 IgG and its Fab fragment, each DOTA-conjugated and ¹¹¹In-radiolabeled, in two syngeneic colorectal-cancer mouse models. The study addresses an important translational step - selecting an optimal antibody format for CCR8-directed theranostics.  The experiments are clear and technically sound, yet several aspects require clarification or additional analysis before publication.

  1. The claim that whole IgG is “optimal” would be stronger with absorbed-dose or time-integrated-activity estimates for tumor, marrow, and kidney.  Using the existing SUV/biodistribution data, please provide basic dosimetric projections (MBq·h/g or Gy/MBq) or at least time–activity curves and tumor-to-organ ratios over time. – Response: We have now generated the tumor to organ ratios and time activities curves for the whole IgG and Fab fragments and added those graphs into the new Figures 5 and 7. The following text was added to the Results: “. Fig. 5 displays the tumor to organ ratios for 111In-IgG (Fig.5A) and 111In-Fab (Fig.5B). The tumor to organ ratios which exceeded value of 1 were observed for kidneys and heart for 111In-IgG and for heart only for 111In-Fab. 111In-IgG time activity curves revealed steady decrease in activity in major organs starting from the 2 hour time point and accumulation of activity in the tumor up to 24 hours (Fig.5A) while 111In-Fab activity peaked in the kidneys at 4 hours, steadily descreased in liver and heart and stayed almost constant in the tumor with slight maximum at 24 hours after which it started to decrease (Fig. 5B).   ……. Fig. 7 displays the tumor to organ ratios for 111In-IgG (Fig.7A) and 111In-Fab (Fig.7B). The tumor to organ ratios which exceeded value of 1 were observed for kidneys and heart for 111In-IgG and for heart and liver - for 111In-Fab while the time activity curves for  111In-IgG and 111In-Fab were quite similar to those in MC38 model. “

  1. It'd be better to report exact injected protein mass (µg or pmol) for both formats.  Because Fab and IgG differ markedly in molecular weight and Fc-mediated recycling, matching by radioactivity alone can misrepresent molar equivalence. – Response: We have added the following information to the Materials and Methods: “Anti-CCR8 111In-IgG was labeled at a 5:1 µCi:µg specific activity whereas the anti-CCR8 111In-Fab was labeled at a 15:1 µCi:µg specific activity to ensure that an equimolar amount of antibody and Fab were injected (40 µg antibody and 13.3 µg Fab).”

  2. Given the persistent Fab renal uptake, it would be valuable to test or cite data for coinfusion with lysine/gelofusine or a cleavable linker.  Alternatively, re-frame the conclusion as “current Fab format limited by kidney retention.” – Response: We have added the following information to the Discussion: “Potential solution to this problem of high kidney uptake could be implementation of linkers cleavable by the renal brush border membrane enzymes which could decrease the radioactive doses to the kidney without impairing the tumor uptake [19]. In addition, positively charged aminoacids such as arginine or lysine were utilized in the past [18] while more recent approaches include co-administration of sodium paraaminohippurate with various radiopharmaceuticals to reduce their kidney uptake [20].”

  3. Add scale bars and color-scale legends to SPECT/CT images; ensure consistent tumor ROI definition. Plot %ID/g and tumor-to-organ ratios alongside SUVs for clarity. – Response: The color scale bars are already present in Figures 4 and 6 and the color scale is also explained in figure legends. We have now plotted the tumor to organ and time activity curves and they are shown in new Figures 5 and 7.

  4. Please compare these findings with other format-comparison studies (e.g., HER2-targeted IgG vs Fab RIT) to place the work in a broader theranostic context in discussion.
    - Response: The comparison has been made with the recent study by Konde et al targeting HER2. The text in the Discussion reads: “In this regard, Kondo et al. have recently reported generation and evaluation of the 225Ac-labeled trastuzumab IgG, F(ab')2 and Fab for theranostic SPECT/CT imaging and alpha-particle radioimmunotherapy of HER2-positive human breast cancer [11]. “

  1. Correct typographical errors (“highes activity” → highest activity; “immune chedck point” → checkpoint; “radiolabelieng” → radiolabeling). – Response: Have been corrected.

  2. Improve figure font size and label legibility, like Figure 1A, Figure 3. – Response: Have been corrected.

Round 2

Reviewer 1 Report

Comments and Suggestions for Authors

The in vivo efficacy evaluation, and the in vitro cytotoxicity assays must be provided

Author Response

The in vivo efficacy evaluation, and the in vitro cytotoxicity assays must be provided. - Response:  We appreciate that the Reviewer would  like to see the results of the in vitro and in vivo efficacy studies. In addition to our response to the Reviewer in the 1st round on why this molecular characterization manuscript does not include efficacy data - we would like to add that: 1) The efficacy of 225Ac-labeled anti-CCR8 antibody against tumor infiltrating Tregs has been already demonstrated in our recent publication Frank C et al Front Immunol 2025 (ref. 7 in the manuscript). 2) The next step needs to be a pre-clinical study where radioimmunotherapy targeting CCR8+ Tregs will be followed by combination of anti-CTLA4 and anti-PD1 immunotherapy which are used in combination in the clinical setting. However, the current worldwide shortage of 225Ac which has considerably worsened in comparison when 3 years ago we began our studies reported in Front Immunol has made us to rethink which radionuclide we need to go into the clinic to ensure its seamless supply for the clinical trials. This is why we are currently working on radiation dosimetry modeling using the imaging data presented in the current manuscript to model the doses which would be delivered to colorectal tumors in patients if the anti-CCR8 antibody would be labeled with such therapeutic radionuclides as 177Lu, 161Tb, 188Re and 67Cu in comparison with 225Ac. These beta emitters have much better availability worldwide than 225Ac. Once the best beta radionuclide is selected from this dosimetry modeling – we will perform and report the efficacy studies which the Reviewer is requesting.